# Precarious Employment and Increased Incidence of Musculoskeletal Pain among Wage Workers in Korea: A Cross-Sectional Study

**DOI:** 10.3390/ijerph18126299

**Published:** 2021-06-10

**Authors:** Sungjin Park, June-Hee Lee

**Affiliations:** 1Department of Occupational and Environmental Medicine, Cheonan Medical Center, Cheonan 31151, Korea; psjin9318@gmail.com; 2Department of Occupational and Environmental Medicine, Soonchunhyang University College of Medicine, Seoul 04401, Korea

**Keywords:** musculoskeletal disorders, employees, non-standard employment, working conditions, occupational health

## Abstract

The number of precarious workers is increasing globally, and precarious employment is becoming a public concern in terms of workers’ health. However, sufficient research on precarious employment and its impact on musculoskeletal pain (MSP) is lacking. This study aimed to investigate the relationship between precarious employment and the risk of MSP among Korean wage workers. After merging the data from the 4th and 5th Korean Working Conditions Surveys, 59,644 wage workers were analyzed. The control group comprised full-time permanent workers, and precarious employment was defined as workers involved in temporary or daily employment, or part-time workers. The outcome variable was the summed number of MSP in three anatomical sites (back, neck and upper limb, lower limb). Zero-inflated negative binomial analyses were selected to estimate the odds ratios (ORs) and 95% confidence intervals (CIs) between precarious employment and MSP. In adjusted models with age, sex, educational level, income level, weekly working hours, and occupation, precarious employment was significantly associated with an increased risk of both MSP (OR 1.66 95% CI 1.56–1.77) and work-related MSP (OR 1.18 95% CI 1.11–1.25). Given the job insecurity and health inequity associated with precarious employment, special attention on precarious workers’ health is needed.

## 1. Introduction

Precarious employment is a collective term that refers to a set of labor market working conditions that are disadvantageous to workers [1]. Although there is no overall consensus regarding a precise definition of the term, precarious employment is often considered to be characterized by temporary employment, income insufficiency, and lack of social protection or worker rights [2,3,4]. Economic globalization, which emerged in the late 1900s, led to intense changes in the labor market seeking more flexible employment [5]. This accelerated the transition from standard employment, characterized by permanent and full-time contracts, to non-standard or atypical types of employment [6]. During the last few decades, the number of workers in precarious employment has grown in many developed countries, including the US, Europe, and Korea [7,8]. In 2019, the combined proportion of daily and temporary employment was estimated to be 21.7% among the economically active population in Korea [9].

Precarious employment is associated with the ill health of workers because of its poor working conditions. The health-related outcomes of precarious employment are mainly the effects on general [10,11,12] and mental health [13,14,15,16,17] and physical injuries [18]. However, the relationship between precarious employment and physical complaints such as musculoskeletal pain (MSP) has received relatively less attention. 

MSP can have direct physical or psychological impacts on individuals [19]. In both developed and developing countries, MSP is a fairly common cause of disability [20] and is one of the most common work-related medical conditions [21]. Moreover, MSP in two or more anatomical sites is not uncommon and is more severe than MSP in single sites [21].

Therefore, more evidence regarding the association between precarious employment and the risk of increased incidence of MSP is required. This study aimed to investigate the relationship, using large-scale national data representing Korean workers. Particularly, the analysis focused on the association between the risk of increased incidence of MSP in multiple body parts and precarious employment. Further, we specifically considered three individual sites—back, the neck and upper limbs, and lower limbs. 

## 2. Methods

### 2.1. Study Design and Participants

This cross-sectional study is based on a secondary analysis of data from the 4th (2014) and 5th (2017) Korean Working Conditions Survey (KWCS) conducted by the Korean Occupational Safety and Health Agency. KWCS gathers information on work environments, exposure to risk factors for workers’ health, and health outcomes. The survey population is representative of economically active working adults, and thus, retirees, the unemployed, housewives, and students were excluded from the study sample. Participants were extracted from multistage, stratified, random sampling from the census and interviewed face-to-face. The reliability and validity of the KWCS data have been confirmed in previous studies [22].

Initially, merging the 4th and 5th KWCS produced the data of 100,212 workers in total, of whom 60,859 were wage workers. After excluding 302 subjects with missing MSP values and 913 with no information on the type of employment, 59,644 wage workers were included in the final analysis.

### 2.2. Measurements

#### Precarious Employment

Participants responded to questions regarding the type of employment (permanent, temporary, or daily) and work schedule (full-time or part-time) associated with their current job. Based on these two criteria, precarious employment was defined as either having temporary or daily employment or working part-time. Workers not falling into this category were regarded as full-time permanent workers.

### 2.3. Musculoskeletal Pain and Work-Related MSP

MSP and work-related MSP (or WMSP) were defined by two questions regarding the prevalence of MSP and whether they were related to work. First, participants responded to self-reported questions about MSP in their body parts such as the back, upper limbs and neck, and lower limbs during the last year. If they had any MSP, they answered further, whether they perceived it as work-related. Additionally, the summed number of MSP in the three body parts was produced as an outcome variable.

### 2.4. Covariates

The models of this study included a range of covariates, including sex, age (20–29, 30–39, 40–49, 50–59, 60 or older), educational level (under high school, high school, above high school), income level (less than 2 million Korean Won, 2 million Korean Won or more), working hours per week (40 or less, 41–52, 53–60, more than 60), occupations, and ergonomic risk factors at the workplace. The occupations were classified into office work (management, professional, or clerical work), service or sales work, and manual work (skilled work related to agriculture, forestry, and fisheries; craft and trade work; equipment, machine operating, and assembling or elementary work). For ergonomic risk factors, exposure for more than half of the working hours to any of the following five risk factors was defined as having exposure: fatigue-inducing or painful posture; lifting or moving people; dragging, pushing, or moving heavy objects; a standing posture; and repetitive hand or arm movements. 

### 2.5. Statistical Analysis

Chi-square tests were conducted to understand the relationship between precarious employment, demographic, socioeconomic, and occupational characteristics and MSP of employees. Both MSP and WMSP are count variables with a range of 0–3 and a skewed distribution to zero, giving them a greater variance than the average value. Zero-inflated negative binomial regression (ZINBR) was selected as the optimum model after comparing the goodness-of-fit of Poisson regression, negative binomial regression (NBR), and ZINBR, which are suitable analytical methods for these variables. The association between precarious employment and the risk of increased incidence of MSP or WMSP was investigated using ZINBR. All analyses were performed using SPSS 26 (IBM Corp, Armonk, NY, USA), and *p*-values < 0.05 were considered statistically significant. 

## 3. Results

Table 1 shows the demographic, socio-economic, and work-related characteristics of the study participants. Approximately a quarter (27.29%) were working in precarious employment. All variables such as age group, sex, educational and income level, weekly working hours, occupation, and ergonomic risk factors were significantly associated with precarious employment. Compared to full-time permanent workers, precarious workers were primarily women, aged 50 or older, had lower levels of education and income, and were engaged in manual work than office work. The proportion of being exposed to five harmful ergonomic risk factors was remarkably higher in precarious workers than in full-time permanent workers (fatigue-inducing or painful posture; lifting or moving people, dragging, pushing, or moving heavy objects, standing posture, repetitive hand or arm movements).

Table 2 shows the significant relationship between precarious employment and MSP (WMSP). The proportion of having any MSP, having two or three MSP in multiple sites, and the average numbers of MSP were significantly higher in precarious workers than in full-time permanent workers. This result was the same for WMSP. Precarious workers had a significantly higher prevalence of back pain, neck and upper limb pain, and lower limb pain than full-time permanent workers (data not shown).

Table 3 shows the odds ratios (ORs) and 95% confidence intervals (CIs) from the ZIBNR analysis. After adjusting for age group, sex, educational level, income level, working hours, and occupation, the risk of increased incidence of MSP was significantly higher in precarious workers than full-time permanent workers (OR 1.66 95% CI 1.56–1.77). Although the magnitude of risk was slightly lower, the OR of increased incidence of WMSP also showed a significant increase in precarious employment (OR 1.18 95% CI 1.11–1.25).

## 4. Discussion

This study identified findings based on nationally representative data with large samples of Korean wage workers. The risk of an increase in the incidence of MSP as well as WMSP was significantly elevated in precarious workers compared to full-time permanent workers. 

Our results are consistent with that of several previous studies. In studies using only the 5th KWCS, the risk of MSP such as back pain, upper limb pain, and lower limb pain was significantly higher among daily workers than standard workers [5]; it was also higher among nonstandard males who are unskilled manual workers, compared to other occupational categories [17]. Moreover, two recent studies which measured multi-dimensional aspects of precarious employment also suggested that precarious workers had an increased risk of having MSP in more than one area [23], and increased risk of back pain, upper limb pain, and lower limb pain [24]. However, in a study using the same methods of measuring multiple dimensions of precarious employment [25], no significant association was observed between precarious employment and MSP. A plausible reason for this inconsistent result could be that the control group in that study was a low precarious group, whereas the control group in the other studies comprised regular or full-time permanent workers. Another reason suggested by the researchers might be that the workers in the highly precarious group were younger than those in the less precarious group and presumably more resistant when it comes to MSP.

Several explanations have been provided for the relationship between precarious employment and health inequity [26,27]. Direct and psychological effects stem from an uncertain future, unfairness, and the powerlessness of precarious employment. Contrastingly, indirect causes include the lack of material or social resources and more chances to be exposed to harmful physical or psychological working conditions. 

Similarly, various hypotheses can be put forward about the relationship between precarious employment and MSP. The results of this study revealed that precarious workers tend to be older, manual workers with greater exposure to ergonomically inappropriate working factors [5,21]. Aging decreases physical ability, flexibility, and muscle strength, making the body vulnerable to injuries, which can increase physical complaints such as MSP [28]. Additionally, social or material deprivation due to precarious employment can restrict the resources required for physical activity [29,30], contributing to lack of exercise and reduction in muscle strength, which is a risk factor for MSP [31,32].

This study is not without its limitations. First, caution must be exercised when establishing causality because of the cross-sectional study design. Researchers have proposed the possibility of reverse causation between precarious employment and ill health, namely the “healthy hire effect”—a hypothesis that healthier workers are employed and remain in full-time permanent work [25]. However, others have suggested that the negative impact of precarious employment on workers’ health is more reasonable [33]. Further studies based on a sufficient sample size will be needed to establish causality. Second, recent studies on precarious employment and MSP have used multidimensional constructs of precarious employment rather than single indicators, such as types of employment [23,24]. However, the evaluation of precarious employment using a type of employment or contract, such as temporary or part-time work, is a widely used method in epidemiological settings [25]. Additionally, our study defined workers as being in precarious employment if they had either temporary or part-time jobs. In other words, this is an effort to be in line with existing studies, where workers could be classified into the highly precarious group if the score of any one specific dimension of several constructs of precarious employment was high [34]. In the future, detailed characteristics of precarious employment should be measured to study its health effects.

## 5. Conclusions

Despite these drawbacks, this study clearly revealed that precarious employment is associated with the risk of increased incidence of not only MSP in general but also MSP related to occupational causes. Considering the unfairness and health inequity in precarious employment, special attention and efforts to improve workers’ health are necessary.

## Figures and Tables

**Table 1 ijerph-18-06299-t001:** General characteristics and working conditions by precarious employment.

Characteristics	Precarious Employment		*p*-Value
	No [n (%)]	Yes [n (%)]	
Total	43,368 (72.71)	16,276 (27.29)	
Age			
20–29	5112 (11.79)	2734 (16.80)	<0.0001
30–39	12,253 (28.25)	1683 (10.34)	
40–49	13,163 (30.35)	2980 (18.31)	
50–59	9299 (21.44)	3711 (22.80)	
60 or above	3541 (8.17)	5168 (31.75)	
Sex			
Men	23,347 (53.83)	6287 (38.63)	<0.0001
Women	20,021 (46.17)	9989 (61.37)	
Educational level			
Middle school or lower	2454 (5.68)	5093 (31.50)	<0.0001
High school	14,145 (32.73)	7571 (46.82)	
College or higher	26,619 (61.59)	3506 (21.68)	
Income level			<0.0001
≥2,000,000 Won	31,179 (72.22)	5238 (32.31)	
<2,000,000 Won	11,992 (27.78)	10,976 (67.69)	
Weekly working hours			
≤40	23,000 (53.03)	10,914 (67.06)	<0.0001
41–52	13,202 (30.44)	2966 (18.22)	
53–60	5230 (12.06)	1568 (9.63)	
>60	1936 (4.46)	828 (5.09)	
Occupation			
Office workers	21,711 (50.28)	1984 (12.21)	<0.0001
Service workers	10,694 (24.76)	6059 (37.28)	
Manual workers	10,778 (24.96)	8210 (50.51)	
Exposure to risk factors for more than half of working time			
Fatigue-inducing or painful posture	11,786 (27.22)	6415 (39.54)	<0.0001
Lifting or moving people	2260 (5.23)	1089 (6.72)	<0.0001
Dragging, pushing, or moving heavy objects	5985 (13.84)	3695 (22.77)	<0.0001
Standing posture	17,630 (40.76)	10,011 (61.75)	<0.0001
Repetitive hand or arm movements	22,324 (51.66)	9841 (60.70)	<0.0001

All *p*-values are produced from chi-square tests conducted on the 4–5th KWCS data.

**Table 2 ijerph-18-06299-t002:** Number of musculoskeletal pain by precarious employment.

Characteristics	Precarious Employment		*p*-Value
	No [n (%)]	Yes [n (%)]	
Number of musculoskeletal pain			
0	29,937 (69.03)	9135 (56.13)	<0.0001
1	6636 (15.30)	2520 (15.48)	
2	4682 (10.80)	2780 (17.08)	
3	2113 (4.87)	1841 (11.31)	
Number of work-related musculoskeletal pain			
0	32,048 (73.90)	10,820 (66.48)	<0.0001
1	7112 (16.18)	2089 (12.38)	
2	5082 (11.56)	2112 (12.98)	
3	1934 (4.4)	1255 (7.71)	
Average number of musculoskeletal pain *	0.51 ± 0.87	0.83 ± 1.08	<0.0001
Average number of work-related musculoskeletal pain *	0.43 ± 0.82	0.62 ± 0.98	<0.0001

***** Mean ± standard deviation.

**Table 3 ijerph-18-06299-t003:** Increased incidence of musculoskeletal pain and precarious employment from the zero-inflated negative binomial regression analyses.

Outcome	Crude	Adjusted ^a^
	OR (95% CI)	OR (95% CI)
Increased incidence of musculoskeletal pain		
Precarious employment		
No	1.00 (reference)	1.00 (reference)
Yes	2.09 (1.98–2.20)	1.66 (1.56–1.77)
Increased incidence of work-related musculoskeletal pain		
Precarious employment		
No	1.00 (reference)	1.00 (reference)
Yes	1.55 (1.47–1.62)	1.18 (1.11–1.25)

OR, odds ratio; CI, confidence interval. ^a^ Adjusted for age group, sex, educational level, income level, weekly working hours, and occupation.

## Data Availability

KWCS data can be found and downloaded publicly at the following link: https://oshri.kosha.or.kr/eoshri/resources/KWCSDownload.do.

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
