# Peer review of "Precarious Employment and Increased Incidence of Musculoskeletal Pain among Wage Workers in Korea: A Cross-Sectional Study"

_ijerph, 2021, doi:10.3390/ijerph18126299_

Round 1

Reviewer 1 Report

Why did the authors put "Key Messages" in this article? What are these? This is not in line with the journal's format, so it should be removed, and if the authors consider this information relevant to the article, they should see how to include it in the "Introduction" of the article.

This article lacks a "Literature Review" to help scientifically frame this article and show the relevance and importance of this publication. With only a few references in the "Introduction", it is not clear how important and relevant this article is.

The information in "Table 1" is very extensive (2 pages), and the information placed at the beginning is very vague. The authors should work on this information and analyze which information is relevant, and perhaps make some graphs so that the information is easier to interpret by the scientific community.

The information in the tables: "Table 1"; "Table 2"; "Table 3", is very extensive (Table 1"; "Table 2"), and the information placed in the beginning of the tables is very vague. There is no justification for the text, nor an analysis and discussion of the results obtained? It is the readers who have to do the analysis and draw conclusions??? I don't think it should be...

The authors should work the information of the tables, and analyze which is the relevant information, and maybe make some graphics so that the information is easier to interpret by the scientific community.

The "Discussion" is very vague and does not discuss the results obtained. More statistical studies could have been done on the data obtained, which would allow other conclusions to be drawn.

This article lacks a "Conclusion". What was the conclusion that the authors made? What are the suggestions that the authors left for future research work?

The "References" are incorrectly formatted. Authors should be careful and review this situation.

The "Authors' Contribution" is not correctly formatted. It should be revised.

Reviewer 2 Report

Dear Authors, 

I would like to congratulate you for your efforts. Language of the study is clear, and manuscript is well written. I have assessed your cross-sectional study with the STROBE guidelines and can say that the study is well organised. Additionally, findings of the study were stated clearly. However, I have some concerns especially on the rationale of the study, importance of the findings for the current literature, and discussion. 

First of all, in the literature there are many studies that suggest the association between precarious employment (non-standard employment) and chronic musculoskeletal pain and chronic musculoskeletal pain conditions.  Also, this study only repeats a well-known association between MSP and precarious model in adjusted models (with age, gender, education level, occupation). Here are some examples of the studies that suggest same findings:

1-) Ahn, J., Kim, N. S., Lee, B. K., Park, J., & Kim, Y. (2019). Non-standard workers have poorer physical and mental health than standard workers. Journal of occupational and environmental medicine61(10), e413-e421.

2-) Matilla-Santander, N., González-Marrón, A., Martín-Sánchez, J. C., Lidón-Moyano, C., Cartanyà-Hueso, À., & Martínez-Sánchez, J. M. (2020). Precarious employment and health-related outcomes in the European Union: a cross-sectional study. Critical Public Health30(4), 429-440.

3-) Park, J., Han, B., Park, J. S., Park, E. J., & Kim, Y. (2019). Nonstandard workers and differential occupational safety and health vulnerabilities. American journal of industrial medicine62(8), 701-715.

4-) Simões, M. R. L., Souza, C., Alcantara, M. A. D., & Assunção, A. Á. (2019). Precarious working conditions and health of metropolitan bus drivers and conductors in Minas Gerais, Brazil. American journal of industrial medicine62(11), 996-1006.

5-) Park, J., & Kim, Y. (2020). Factors Related to Physical and Mental Health in Workers With Different Categories of Employment. Journal of occupational and environmental medicine62(7), 511-518.

6-) Park, S., Lee, J., & Lee, J. H. (2021). Insufficient Rest Breaks at Workplace and Musculoskeletal Disorders Among Korean Kitchen Workers. Safety and Health at Work.

In the discussion, you have only mentioned only two studies that find similar findings. However, the discussion is lacking how these studies differ from your research. Therefore, it is important to also discuss similarities and differences of the methods, scope and focus of the studies. Because it seems that you only repeat the existing studies in the literature.

In your methods you classify musculoskeletal pain into two categories. First one is musculoskeletal pain and second one is work related musculoskeletal pain. Pain is a subjective feeling and depends on preson's own experience and verbal expression. Thus, it is appropriate to ask participants whether they had pain or not during last year. However, to associate this pain with the occupation with a verbal expression is prone to bias and is an inappropriate approach.

Lastly, I would suggest writing a separate conclusion rather than embedding it into the discussion.

Kind Regards,

Omer Elma

Reviewer 3 Report

I have had the pleasure of reviewing the manuscript entitled "Precarious employment and musculoskeletal pain in wage workers in Korea". The authors set out to investigate if a relationship between precarious employment and musculoskeletal disorders exists. To do so they use data from a Korean census. The study is well written and very clear in its statements. I would like to congratulate the authors on that.

I do have some comments for you to consider:

Title: I suggest that you add: “…workers in Korea: a cross-sectional study” to the title.

Lines 23-24: The authors request policies to reduce employment insecurities or improve the physical activity of precarious workers. Does the KWCS data offer measurements of the qualities of the study populations’ level of physical activity or level of employment security?

Line 62-67: Please state the aim of the study explicitly.

Lines 88-89: Although discussing the definition of precarious employment above, the authors finally only differentiate on working hours and not on income, social protection, or workers’ rights. Could the results have been different had the criteria for precarious employment in the study population been different?
Could some included study participants be working less than full time without being characterized by any of the above?

Lines 92-98: Does the KWCS data offer any magnitude of the measured variables of pain? Would it be relevant to include the intensity of which the study population experienced pain instead of using it binomially?

Line 133: “…, and engaged in manual work than office work”. Is something missing in this sentence? It seems unclear. Please correct or rephrase.

Table 1: This table should be compressed significantly. Also, what do the p-values represent? What tests were carried out on what data to come up with these p-values? And in the age-section there is only one p-value. What does that represent?

Lines 138-139 and Table 2: In table 2 you present the results of a series of hypothesis tests. This has nothing to do with a statistical relationship. You present differences in subgroups by precarious employment. Please rephrase the text to match the statistical test.

Lines 199-202: Please conclude properly on your results based on the drawbacks you have stated in the discussion. Also, please refrain from making political statements. I suggest you strike “unfairness” and rephrase the sentence.

Round 2

Reviewer 1 Report

 - Page 8 of the article should be removed because it is blank and has no text;

- The article's References continue with the text badly formatted. I recommend that the authors consult other articles of the journal already published.

Reviewer 2 Report

Dear Authors, 

I have checked your response carefully and revised the paper one more time. I am satisfied with the response you have given and the changes you have made in the text. Therefore, I am personally changing my decision from "major revision" to "accept in present form". I congratulate you for this work. 

Kind Regards, 

Omer Elma 

Reviewer 3 Report

I have no further comments